# Effect of Controlled Atmosphere Packaging on the Physiology and Quality of Fresh-Cut *Dictyophora rubrovolvata*

**DOI:** 10.3390/foods12081665

**Published:** 2023-04-17

**Authors:** Ziqian Xia, Rui Wang, Chao Ma, Jiangkuo Li, Jiqing Lei, Ning Ji, Xianxing Pan, Tongjie Chen

**Affiliations:** 1College of Food and Pharmaceutical Engineering, Guiyang University, Guiyang 550000, China; xiaziqian0603@163.com (Z.X.); gyuchaoma@163.com (C.M.); jiqinglei@gmail.com (J.L.); jining552100@163.com (N.J.); pxx991025@163.com (X.P.); 2Tianjin Key Laboratory of Postharvest Physiology and Storage of Agricultural Products, National Engineering and Technology Research Center for Preservation of Agricultural Produce, Tianjin 301699, China; lijkuo@sina.com; 3Gui Zhou Mei Wei Xian Dictyophora Industry Company Limited, Zhijin 552100, China; wo2427713193@163.com

**Keywords:** controlled atmosphere, fresh-cut *D. rubrovolvata*, physiology, nutrition, umami

## Abstract

*Dictyophora rubrovolvata* is a typical edible fungus of Guizhou Province and is very popular due to its unique taste and texture. In this study, the effect of a controlled atmosphere (CA) on fresh-cut *D. rubrovolvata* shelf life was investigated. Firstly, this study addresses the influence of different O_2_ concentrations (5%, 20%, 35%, 50%, 65%, 80%, or 95%) with N_2_ balance on fresh-cut *D. rubrovolvata* quality while stored at 4 ± 1 °C for 7 d. Then, on the basis of the determined O_2_ concentration (5%), CO_2_ (0%, 5%, 10%, 15%, or 20%) was involved and stored for 8 d at 4 ± 1 °C. Evaluations of physiology parameters, texture, browning degree, nutritional, umami, volatile components, and total colony numbers were determined in fresh-cut *D. rubrovolvata*. From the results of water migration, the sample of 5% O_2_/5% CO_2_/90% N_2_ was closer to 0 d than other groups at 8 days. Meanwhile, the polyphenol oxidase (2.26 ± 0.07 U/(g·min)), and catalase activity (4.66 ± 0.08 U/(g·min·FW)) were superior to the samples of other treatment groups on the eighth day (3.04 ± 0.06 to 3.84 ± 0.10 U/(g·min), 4.02 ± 0.07 to 4.07 ± 0.07 U/(g·min·FW)). Therefore, we found that a gas environment with 5% O_2_/5% CO_2_/90% N_2_ could ensure the membrane integrity, oxidation, and prevent the browning of fresh-cut *D. rubrovolvata*, thus better maintaining the physiological parameters. Meanwhile, it also maintained the samples’ texture, color, nutritional value, and umami taste. Furthermore, it inhibited the increase in total colony numbers. The volatile components were closer to the initial level compared with other groups. The results indicate that fresh-cut *D. rubrovolvata* could maintain its shelf life and quality when stored in 5% O_2_/5% CO_2_/90% N_2_ at 4 ± 1 °C.

## 1. Introduction

Edible fungi have become an important part of the human diet due to having a high quality of protein, vitamins, minerals, low calorie content, and other components [1]. Vegetarians are also fond of them because of their essential amino acids from animal protein and natural sources of vitamin D_2_ [2]. Consequently, the production and consumption have increased significantly as they are incorporated into consumers’ diets. China is one of the world’s largest producers and exporters of edible fungi [3]. In terms of value, the mushroom industry ranks after grain, vegetable, fruit, and edible oil plantations, and is higher than sugar, cotton, and tobacco.

*D. rubrovolvata*, which belongs to the *Phallaceae* family and roots in the forest or by broad-leaved trees on the ground, has been used for centuries as a food and medicine [4,5,6]. It is highly nutritious and rich in polysaccharides, polyphenols, flavonoids, and other bioactive substances. It improves immunity; has anticancer, antioxidant, and anti-inflammatory medicinal properties; and is deeply valued by consumers in China [7]. It was planted in 1983 [6]. As a result of its unique environmental conditions and climate conditions, Guizhou is an ideal place for *D. rubrovolvata* to grow. Meanwhile, because it provides high nutritional, medical, and economic value, *D. rubrovolvata* is currently one of the most popular edible fungi grown commercially in Guizhou Province, China, and is widely cultivated [5]. As a key edible fungus in Guizhou Province during the 13th Five-Year-Plan period, *D. rubrovolvata* played an important role in providing employment, stimulating entrepreneurship, and improving the economic level. In 2021, the annual output of *D. rubrovolvata* in Guizhou Province reached 8800 tons. At present, it is cultivated on a large scale in Zhijin County, Annong County, and other places in Guizhou Province. However, the fruiting body of *D. rubrovolvata* will soften and decay rapidly after harvesting. Because of the high respiration rate, significant water loss, and susceptibility to microorganisms during shelf life, there is a shorter shelf life for edible fungi than for fruits and vegetables [8]. Therefore, post-harvest treatment will become important. Furthermore, there are no reports related to the preservation of *D. rubrovolvata*.

In recent years, as consumers’ awareness of health has increased, more and more people have enjoyed fresh edible fungi, resulting in increased production and consumption. Since fresh-cut edible fungi contain many nutritional ingredients such as vitamins, minerals, and bio-active compounds, they have become extremely popular among consumers [9]. Generally, the shelf life of an edible fungus under normal temperature is 1–3 days [10]. Following cutting, the shelf life decreases due to increased respiration rates, which may result in the consumption of nutrient substances. As a result of mechanical damage during processing, there are a number of physiological and biochemical changes that occur in edible fungi, such as browning, nutrient loss, water loss, and microbial pollution, which would shorten their shelf life and, therefore, affect sales and consumption. The deterioration rate of fresh-cut products is much higher than that of unprocessed ones. Therefore, an effective storage and preservation technology is essential for this emerging industry, but a huge challenge.

In order to prolong the shelf life of fresh-cut fruits, vegetables, and edible fungi, inhibit microorganisms, and maintain product quality, controlled atmosphere (CA) packaging has been studied and employed. Notably, a CA has been widely accepted by manufacturers and consumers because this technology extends the seasonable availability of produce, maintains the physicochemical and functional quality, and can reduce consumer costs [11]. Furthermore, the use of natural gas ingredients guarantees food safety. As a result, the processed products are free of toxic residues. In the beginning, the application of a CA generally consists of decreasing O_2_ and increasing CO_2_. Amodio et al. [12] confirmed that a CA of 3% O_2_/20% CO_2_/77% N_2_ could effectively slow down browning and hardness decline and inhibit the growth of mold in fresh-cut *Pleurotus eryngii*. A study found that fresh-cut figs stored in a CA of 3% O_2_/18% CO_2_/79% N_2_ had significantly lower respiration rates, weight loss, and microbial growth than those stored under normoxic conditions [13]. In Esmaeili et al.’s study [14], 5% O_2_/10% CO_2_/85% N_2_ bound to the aloe vera concentrated gel could effectively maintain the anthocyanin and vitamin C content and inhibit the growth of microorganisms in strawberries. Recently, a CA with high O_2_ has also been proven effective for the produce. According to Li et al. [15], a CA of 80% O_2_/ 20% CO_2_ increased the shelf life of *Agaricus bisporus* by retaining higher energy, suppressing ROS levels, and enhancing enzymatic activity. Another study found that CA treatment of 80% O_2_/20% CO_2_ inhibited polyphenol oxidase (PPO) and delayed browning of *Agaricus bisporus* pericarp and flesh. Additionally, it was able to maintain the integrity of cell membranes and provided good sensory properties [16]. A study found that high O_2_ and CO_2_ (80% O_2_, 20% CO_2_) treatment would improve wound healing and maintain quality by enhancing the activity of antioxidant enzymes, including SOD, CAT, APX, and GR, thus reducing water loss and browning for fresh-cut *Agaricus bisporus* [17,18]. It was in no doubt that CA packaging proved to be effective for preserving edible fungi, fruit, and vegetables for a long time.

The color and the texture were the main factors that influenced consumers’ favorites for edible fungi, which would affect the product value. We found that the tissues of fresh-cut *D. rubrovolvata* were exposed to the environment due to cutting, leading to faster decay and browning. The resulting decay and browning severely limited the consumption of *D. rubrovolvata*. However, so far, there is no available literature on the storage of fresh-cut *D. rubrovolvata* under a CA. Therefore, the present study was performed according to a two-step screening. Firstly, the effect of O_2_ concentration on the fresh-cut *D. rubrovolvata* was investigated. Then, on the basis of the O_2_ concentration being determined, CO_2_ was involved. A series of physiological and biochemical changes in fresh-cut *D. rubrovolvata* under CA conditions was investigated. Through the study of a series of physiological and biochemical color, texture, umami, flavor compounds, and other indicators, the best storage conditions of fresh-cut *D. rubrovolvata* were determined.

## 2. Materials and Methods

### 2.1. Materials

*D. rubrovolvata* was collected from a farm of Anlong Count (Guizhou Province, China, UTMX: 104°59′–105°41′ UTMY: 24°55′–25°33′). The fruiting bodies were harvested on 7 September and 17 September 2021, respectively. Afterward, samples were transported to the laboratory within 3 h after harvest. The polyamide package was purchased from Shijiazhuang Xilong Packaging Co., Ltd. (Shijiazhuang, China), the hydrogen peroxide, ethanol absolute, sodium bicarbonate, etc., from China National Medicines Co., Ltd. (Beijing, China), the amino acid standard from Sigma-Aldrich Trading Co., Ltd. (Shanghai, China), 5′-inosine monophosphate (5′-IMP) standard, 5′-guanosine monophosphate (5′-GMP) standard, 5′-xanthosine monophosphate (5′-XMP) standard and 5′-adenoshine monophosphate (5′-AMP) standard from Standford Chemicals Co. (Lake Forest, CA, USA).

### 2.2. Treatment and Storage

Fruit bodies similar in size (6–8 cm stalk length) and free of pests and mechanical damage were chosen for the experiment. After the fruit body was washed for 30 s in tap water and an electric fan was used to remove the *D. rubrovolvata* surface water, the stipe was cut perpendicularly into three sections and pre-cooled at 4 ± 1 °C for 12 h.

In a preliminary experiment, the pre-cooled fresh-cut *D. rubrovolvata* was packaged into polyamide packages (22 cm × 32 cm, 0.24 mm thickness, moisture permeability 3.12 g/(m^2^ 24 h), oxygen permeability 46 cm^3^/(m^2^ 24 h 0.1 MPa)), with different gas compositions: 5% O_2_, 20% O_2_, 35% O_2_, 50% O_2_, 65% O_2_, 80% O_2_, and 95% O_2_, respectively, and underwent N_2_ balance and were stored at 4 ± 1 °C for seven days. Samples were taken out and analyzed at 0, 3, 5, and 7 d, with 6 bags taken for each treatment to assess the shelf life. Then, a total of 18 pieces of *D. rubrovolvata* were loaded into each bag with a weight of approximately 240 g. Samples were packaged in atmosphere with different gas compositions: 5% oxygen (O_2_), 0% carbon dioxide (CO_2_), 95% nitrogen (N_2_) (5/0/95); 5% O_2_, 5% CO_2_, 90% N_2_ (5/5/90); 5% O_2_, 10% CO_2_, 85% N_2_ (5/10/85); 5% O_2_, 15% CO_2_, 80% N_2_ (5/15/80); and 5% O_2_, 20% CO_2_, 75% N_2_ (5/20/75), respectively, then heat-sealed with a gas packing machine. For simulation of sales scenarios, samples were stored at 4 ± 1 °C during the shelf life. Samples were taken out every two days (0, 2, 4, 6, and 8 days) during the shelf life, with 12 bags taken from each treatment and analyzed.

### 2.3. Decay Rate, Respiration Rate, Malondialdehyde (MDA) Content, Relative Conductivity, Ethanol Content, and Low-Field Nuclear Magnetic Resonance (LF-NMR) Transverse Relaxation Measurements

A decay rate analysis of fresh-cut *D. rubrovolvata* was performed in accordance with Rux et al., [19] with minor modifications. Quality evaluation based on visual observation of fresh-cut *D. rubrovolvata* for each of the CA packaging was performed at the end of storage by 10 panelists. The fruiting bodies with a surface browning and generating the off-flavor were identified as decay. The ratio of decayed samples to the total number of samples analyzed was stated as the decay rate (%).

The respiration rate of the fruiting body was determined by Zhang’s method, with some modifications [20]. Before measurement, 120 g of fruiting bodies of *D. rubrovolvata* were placed in 1 L sealed plastic jars at 25 °C for 3 h. To analyze gas samples from the headspace of the plastic jars, gas syringes were used to draw 2 mL of gas from each and analyze them with infrared O_2_ and CO_2_ analyzers (Checkpoint II Portable residual oxygen meter, Dansensor, Rinsted, Denmark). In this study, the respiratory rate was expressed as mg/kg/h CO_2_.

Huang’s method was used to assay the MDA content, with slight modifications [21]. *D. rubrovolvata* (0.2 g, FW) was ground and homogenized in 12 mL of 10% (*w*/*v*) trichloroacetic acid (TCA) and filtered to obtain a sample extract. We added 5 mL filtrate and 5 mL 0.6% thiobarbituric acid (TBA) to the tube. The solution was vibrated and mixed before it was boiled for 15 min and cooled to room temperature. A final measurement was made of the supernatant’s absorbance at 450 nm, 532 nm, and 600 nm. The MDA content of the fresh sample weight was measured as mmol/g.

Based on He et al.’s method [22], the relative conductivity was analyzed in the fruiting body, with a slight modification. A total of 40 mL of deionized water was added to the triangular flask its conductivity (P_0_) was measured. The fruiting body of *D. rubrovolvata* was cut, mixed (2 × 2 mm) and transferred into triangular flasks. A conductivity meter was used to measure the conductivity of deionized water and fresh samples (P_1_). It was oscillated for 10 min and stood still at 25 °C. Triangular flasks were cooled to room temperature (about 25 °C) after being placed in boiling water (100 °C) for 10 min and their conductivity (P_2_) was measured. The relative conductivity was expressed as a percentage.
$$\mathrm{Relative}\;\mathrm{conductivity}\;\left(\%\right)=P_{1}-P_{0}/\left(P_{2}-P_{0}\right)\times 100\%$$

According to Ventura-Aguilar’s description, headspace gas chromatography was used to determine *D. rubrovolvata*’s ethanol concentration [23]. Results were expressed as μg/g fresh weight.

An LF-NMR measurement was performed using a method described by Cheng et al. [24].

### 2.4. Texture, Browning Degree, Polyphenol Oxidase (PPO) Activity, Catalase (CAT) Activity, and Total Phenolic (TP) Content

According to Jahanbakhshi et al.’s report, the shear force of the fruiting body was determined [25].

Using Liu et al.’s method with minor modifications, the browning degree of fruiting bodies was measured [26]. A mixture of approximately 2 g of tissue was ground in an ice bath with 8 mL of 0.2 mol/L sodium phosphate buffer (pH 6.8) and stored at 4 °C for 15 min, and the mixture was centrifuged at 24,200× *g*. A UV-vis spectrophotometer was used to measure the 420 nm absorbance (Cary 60, Agilent, Santa Clara, CA, USA). The browning degree was 420 nm.

In order to determine the PPO activity, Liu et al.’s method was used [26]. The PPO activity was calculated as U/(g·min) fresh weight.

A modified method was used to determine the CAT activity [18]. After 2 g of powdered fruiting bodies was homogenized with 3.2 mL phosphate buffer (0.05 mol/L, pH 7.0), it was centrifuged at 24,200× *g* and 4 °C for 10 min. Next, 1.6 mL Tris-HCl solution and 2.72 mL deionized water were added to the test tube with 1 mL supernatant. After immersion in 37 °C water for 3 min, the mixture was transferred into a colorimetric dish, with 0.4 mL of hydrogen peroxide solution. Every 30 s for 3 min, spectrophotometric measurements at 240 nm were taken to determine the absorbance. CAT activity is expressed in units of U/(g·min·FW).

Using a Folin–Ciocalteu assay, the total phenolic (TP) content was determined [27]. The total phenolic content of the sample was calculated using a gallic acid standard curve and expressed in mg of gallic acid equivalent (GAE)/g of the sample.

### 2.5. Polysaccharides Content

Polysaccharides were determined using a method described by Wang et al. [28]. The polysaccharides content was expressed as g/(100 g).

### 2.6. Free Amino Acid

According to Wang et al.’s study [29], the free amino acid content was detected for fruiting bodies. Initially, 0.5 g of fresh samples were crushed and extracted with 5 mL of 0.01 mol/L hydrochloric acid. The sample was then heated in a boiling water bath for 30 min, followed by 10 min of centrifugation at 24,200× *g*. After separating the supernatant, we added 2 mL of 0.01 mol/L hydrochloric acid, followed by 5 min of sonication. We centrifuged the mixture, combined it with a supernatant, and added solvent until the volume was 10 mL. Determination was performed using a 0.22 μm filter membrane. Analysis was performed using an Agilent 1100 liquid chromatograph (with VWD detector) (1100, Agilent, Santa Clara, CA, USA) containing a ZORBAX Eclipse AAA column (4.6 × 150 mm, 3.5 μm). The HPLC conditions were as follows: mobile phase A: 40 mmol/L sodium dihydrogen phosphate (pH 7.8); mobile phase B: acetonitrile/methanol/water = 45/45/10; flow rate: 1.0 mL/min; UV detection wavelength, 338 nm (0 to 19 min), 266 nm (19.01 to 25 min); oven temperature: 40 °C; injection volume: 1 μL. Identification and quantification of each amino acid were conducted using a standard mixture of 17 amino acids (Sigma-Aldrich CN Inc., Shanghai, China).

### 2.7. 5′-Nucleotides

Wang et al.’s procedure was used to measure 5′-nucleotides, with minor modifications [29]. In 25 mL deionized water, 0.2 g of mushroom powder was boiled for 5 min, cooled, and centrifuged for 15 min at 24,200× *g*. A second extraction was carried out by adding 25 mL of deionized water after filtration. Prior to HPLC analysis using a RIGOL L-3000 high-performance liquid chromatography system (RIGOL L-3000, Puyuan, Beijing, China) equipped with a Diamonsil C18 column (250 × 4.6 mm, 5 μm), a 0.45 μm cellulose membrane was used to filter the extraction. The HPLC conditions were as follows: mobile phase, 0.01 mol/L KH_2_PO_4_ (pH 4.68); flow rate, 1.0 mL/min; UV detection wavelength, 254 nm; oven temperature, 25 °C; and injection volume, 20 μL. Standards for 5′-nucleotides were used to identify and quantify each nucleotide (Sigma-Aldrich CN Inc., Shanghai, China).

### 2.8. Equivalent Umami Concentration (EUC)

The equivalent umami concentration, or EUC (g MSG/kg), is defined as the MSG concentration corresponding to the umami intensity created by the synergy between umami amino acids and 5′-nucleotides, and calculated as follows:(1)$$Y=\sum \;a_{i}b_{i}+12.18\left(\sum \;a_{i}b_{i}\right)\left(\sum \;a_{j}b_{j}\right),$$
where Y is the EUC value of the mixture in terms of g MSG/kg; ɑ_i_ is the concentration (g/kg) of each umami amino acid (aspartic acid (Asp) or glutamic acid (Glu)); ɑ_j_ is the concentration (g/kg) of each 5′-nucleotides (5′-IMP, 5′-GMP, 5′-XMP, or 5′-AMP); b_i_ is the relative umami concentration (RUC) for each umami amino acids with respect to MSG (Asp = 0.077, Glu = 1); b_j_ is the RUC for each umami 5’-nucleotides constant (5′-GMP = 2.3, 5′-IMP = 1, 5′-XMP = 0.61, 5′-AMP = 0.18); and 12.18 is a synergistic constant.

### 2.9. Electronic Nose

Following the measurement of Pei et al. [30], an electronic nose analysis of fresh-cut *D. rubrovolvata* was conducted.

### 2.10. Total Colony Numbers

As per Shen et al. [31], the total number of colonies was determined by aerobic plate counting. A colony-forming unit (CFU) represents the total number of colonies.

### 2.11. Statistical Analysis

The analyses were repeated three times, and the results were expressed as mean ± standard deviation. IBM SPSS 25 statistical software (Armonk, NY, USA) was used for the data processing and analysis. Origin 2017 was used for mapping and Duncan’s method was used for multiple comparisons.

## 3. Results and Discussion

### 3.1. Effect of O_2_ Concentration (%) on the Shelf Life and Quality of Fresh-Cut D. rubrovolvata

To evaluate the effect of O_2_ concentrations (5%, 20%, 35%, 50%, 65%, 80%, 95%) on the shelf life of *D. rubrovolvata*, the shear force, relative conductivity, and decay rate were employed. According to Figure 1, the relative conductivity, shear force, and decay rate of all samples increased over its shelf life. In the first three days of storage, all three indicators of samples in a high O_2_ (95%, 50%, 65%) environment were close to the initial value. However, with the storage time extended, the three indicators of samples in a low O_2_ environment were significantly lower than those in a high O_2_ environment. On the seventh day, the relative conductivity, shear force, and decay rate of samples under a 5% O_2_ environment were 26.91%, 8.20 ± 0.19 N, and 17.67%, respectively, which were 19.77%, 17.98%, and 33.75% lower, respectively, than ones at 95% O_2_/5% N_2_ (*p* < 0.05). The above result suggested that a low O_2_ concentration (5%) was more efficient at inhibiting the changes in the relative conductivity, shear force, and decay rate during the shelf life of fresh-cut *D. rubrovolvata*. The results of Li and Lyn’s papers confirm the same findings [32,33].

### 3.2. Effect of CO_2_ Concentration (%) with 5% O_2_ on the Shelf Life of Fresh-Cut D. rubrovolvata

#### 3.2.1. Changes in Physiological Parameters

The appropriate concentration of CO_2_ is also very important. An excessive CO_2_ concentration (CO_2_ concentration of more than 12%) could cause physiological damage to edible fungi [32]. At the optimal O_2_ concentration (5%), the effects of CO_2_ concentrations (0%, 5%, 10%, 15%, 20%) were introduced and investigated. Figure 2 illustrates the relationship between shelf life and physiological parameters in fresh-cut *D. rubrovolvata*. Clearly, the higher the concentrations of CO_2_ (10%, 15%, or 20%), the higher the decay rate of the samples (Figure 2a). The highest decay rate in all treated fruiting bodies was observed at 8 d. A sample kept in a CA of 5% O_2_/5% CO_2_/90% N_2_ provided a decay rate of 12.93%; samples under O_2_ group alone treatments and other CO_2_ groups were 1.45 times, 1.76 times, 2.43 times, and 2.03 times higher than that of 5% O_2_/5% CO_2_/90% N_2_, respectively (*p* < 0.05).

The respiration of edible fungus after harvest largely determines the quality changes in the sample. Plant cells use energy reserves more efficiently when their respiration rate increases, resulting in reducing their quality and shelf life [20]. Capotorto et al. [34] showed that the combination of low O_2_ and high CO_2_ (10% O_2_/10% CO_2_/80% N_2_) could effectively inhibit the increase in the respiratory rate of fresh-cut artichokes, thereby maintaining the nutrition and quality of the sample. In our study, the respiration rate of a sample under a 5% O_2_/5% CO_2_/90% N_2_ gas environment was 38.26 ± 1.78 mg CO_2_/kg/h (*p* < 0.05), while the respiration rate of samples under a CO_2_ (0%, 10%, 15%, or 20%) storage environment was 48.14 ± 2.23, 60.16 ± 1.70, 71.32 ± 1.91, and 65.08 ± 1.76 mg CO_2_/kg/h on the eighth day, respectively. The respiratory rate showed a similar trend to the decay rate (Figure 2b); the higher the respiratory rate, the higher the decay rate of the sample.

The MDA and relative conductivity are widely used to evaluate the membrane and tissue integrity of edible fungi, fruits, and vegetables, which are an important product of membrane lipid peroxidation. The fact that the content of the MDA changed is an important indicator of the degree of damage to the membrane system. The higher the content, the more serious the degree of damage to the membrane system [20]. The MDA content in a fruiting body at storage was 9.06 ± 0.38 mmol/g FW for the initial value and obviously increased during shelf life (Figure 2c). The MDA content in all samples was 62.82 to 190.71% higher than that on the initial day, and the minimum value was observed under 5% O_2_/5% CO_2_/90% N_2_ (14.75 ± 0.48 mmol/g FW) conditions after 8 d (*p* < 0.05). Li et al. [32] found that 2% O_2_/30% CO_2_/68% N_2_ could result in better inhibition of MDA accumulation in *Pleurotus eryngii* and it also alleviated lipid and membrane peroxidation in mushrooms to the maximum extent in comparison with other groups. This result could demonstrate that an application of low O_2_ combined with CO_2_ treatment would be a better way to preserve postharvest products. The relative conductivity increased in all groups over time. The increase in relative conductivity indicated a decrease in cell membrane integrity. It was found that fruiting bodies in higher CO_2_ (10%, 15%, or 20%) presented a significantly higher level than 5% O_2_/5% CO_2_/90% N_2_ after 4 d (Figure 2d). On the eighth day, the relative conductivity under CO_2_-treated (10%, 15%, or 20%) fruiting bodies was 38.16 to 59.08% higher than a CA of 5% O_2_/5% CO_2_/90% N_2_ (*p* < 0.05). This might be due to the fact that compared with other groups, 5% O_2_/5% CO_2_/90% N_2_ could well inhibit the respiration rate (Figure 2e) of the sample, thus effectively protecting the integrity of the cell membrane.

As a volatile substance, ethanol can be accumulated during anaerobic conditions, which affects the normal metabolism of fruiting body cells [23]. In the present study, treatments stored under higher CO_2_ were higher in ethanol than those stored under lower CO_2_, and the level of ethanol increased obviously during the shelf life as well (Figure 2e). When stored until the eighth day, the ethanol content of the 5% O_2_/5% CO_2_/90% N_2_ treatment group was the lowest (1882.22 ± 102.42 μg/g), and significantly lower than that of the other treatment groups (3087.84 ± 145.21 to 5394.45 ± 120.67 μg/g) (*p* < 0.05). Similar to our research results, Wang et al. [35] found that a low O_2_ and high CO_2_ (3% O_2_/7% CO_2_/90 N_2_) storage environment could effectively inhibit the accumulation of ethanol and maintain good sensory quality of garlic scapes.

The moisture content and existing state of fresh fruiting bodies were important physical indexes of freshness. Water molecule dynamics and environment during edible fungi storage degradation were assessed by measuring the spin-spin relaxation time (T_2_) of the sample. Figure 2f shows the distribution of transverse proton relaxation data and two proton components in fresh-cut *D. rubrovolvata* samples. Unlike the results of other researchers, where pools of water were attributed to the vacuole, cytoplasm, and cellular wall, only two peaks were observed for water relaxation times in our experiment [24]. It might be due to the destruction of cell tissue caused by cutting, which made the enzyme in different organelles contact the substrate to initiate the wound response induced by the signal. At the same time, due to the destruction of the subcellular structure, the solutes retained in the organelles were released, which were bound to water through hydrogen bonds [35]. The maximum amplitude T_2_ value (95.48 ms) of the sample under the 5% O_2_/5% CO_2_/90% N_2_ environment was found on the eighth day of storage, the same as on day 0. As CO_2_ concentrations increased, the maximum amplitude T_2_ value of the samples continued to change, of which the maximum amplitude T_2_ values of the samples under a CO_2_ (15%, 20%) environment were 77.53 ms and 109.70 ms. Compared with the initial level, they changed by 18.80% and 14.89%. Consistent with the results of MDA and relative conductivity (Figure 2c,d), a 5% CO_2_ gas environment could reduce the oxidative stress reaction caused by mechanical injury and the oxidative damage of the membrane.

Overall, low O_2_ and plausible CO_2_ (5% O_2_/5% CO_2_/90% N_2_) could delay the physiological and biochemical changes of fruiting bodies at 4 °C, inhibit their metabolism, delay the increased of cell membrane permeability, and delay aging, leading to improved storage quality of fresh-cut *D. rubrovolvata*.

#### 3.2.2. Changes in Texture and Browning Degree

Subsequently, we evaluated the fresh-cut *D. rubrovolvata* samples’ texture and browning degree due to the different CO_2_ treatments (0%, 5%, 10%, 15%, or 20%) (Figure 3). The shear force and browning degree during shelf life significantly increased due to a higher CO_2_ concentration. Wang et al. [36] found that with the extension of storage time, under the condition of low O_2_ and high CO_2_, the hardness of the *Pleurotus eryngii* showed an increasing trend, which was consistent with our research. The reason for this phenomenon is that cutting and low-temperature storage increase the lignin content of edible mushrooms, leading to an increase in hardness [36]. The texture is the main attribute responsible for consumers’ acceptance of fresh-cut *D. rubrovolvata*. During the first two days of storage, the shear force of fruiting bodies in each treatment showed no significant change (*p* < 0.05) (Figure 3a). On the eighth day of shelf life, the shear force of the fruiting body in the 5% O_2_/5% CO_2_/90% N_2_ group increased from 5.72 ± 0.53 N at day 0 to 8.79 ± 0.55 N, significantly lower than other storage environments (12.68 ± 0.62 to 16.43 ± 0.75 N) (*p* < 0.05). From the results of the MDA and relative conductivity (Figure 2c,d), we concluded that an appropriate gas environment could inhibit the accumulation of the lignin, and thus the increase in shear force was delayed (Figure 3a).

Among the quality attributes that affect consumers’ purchasing behavior, color is the most intuitive influencing factor. A continual increase in the browning degree (Figure 3b) of the samples was observed during the shelf life. Interestingly, fresh-cut fruiting bodies did not turn brown after storage for 2 d at a CA of 5% O_2_/5% CO_2_/90% N_2_, whereas slight browning was observed in the 5% O_2_/15% CO_2_/80% N_2_ and 5% O_2_/20% CO_2_/75% N_2_ treatment groups. The browning degree of the fruiting body under a CA of 5% O_2_/5% CO_2_/90% N_2_ was lower (4.71 ± 0.13) than in other groups (5.60 ± 0.17 to 7.08 ± 0.18) after storage for eight days (*p* < 0.05). The reason for this phenomenon might be that a high concentration of CO_2_ would cause CO_2_ damage to the fruiting bodies or anaerobic respiration at the later stage of storage, aggravating the fruiting bodies’ browning. Meanwhile, it also increased the relative conductivity and ethanol content (Figure 2d,e).

#### 3.2.3. Changes in Enzyme Activity

As some researchers reported, the oxidative stress reaction of fresh-cut edible fungi, fruits, and vegetables generated from mechanical injury stress would cause membrane lipid peroxidation, damage the integrity of the cell membrane, and lead to a series of changes such as browning and aging [31]. PPO activity plays an important role in the browning reaction of fresh-cut products. Likewise, an antioxidant enzyme such as CAT can represent a plant’s defense capacity. Figure 4 showed the changes in the enzyme activities of PPO and CAT during fresh-cut *D. rubrovolvata* storage at different CO_2_ levels (0%, 5%, 10%, 15%, or 20%). The PPO activity of samples in different storage environments showed an upward trend (Figure 4a). Consistently, the PPO activity of samples stored under 5% O_2_/5% CO_2_/90% N_2_ conditions differed significantly from that of other treatments during the shelf life (*p* < 0.05). After storage for eight days, the PPO activity of a sample in a 5% O_2_/5% CO_2_/90% N_2_ storage environment only increased by 59.64% compared with that at the initial value and was lower than that in other gas environments (82.04 to 129.94%) (*p* < 0.05). Ye et al. [37] found that higher CO_2_ accumulation induced a higher PPO activity in fresh *Lentinula edodes*.

Nevertheless, it is worth noting that the CAT activity of the samples first decreased, then increased, and finally decreased over the shelf life (Figure 4b). Over the course of the shelf life, the CAT activity of samples in a higher CO_2_ storage environment (10%, 15%, or 20%) reached the maximum on the fourth day and then decreased. However, the CAT activity of samples under 5% O_2_/0% CO_2_/95% N_2_ and 5% O_2_/5% CO_2_/90% N_2_ storage environments reached the maximum on the sixth day, and the CAT activity of 5% O_2_/5% CO_2_/90% N_2_ group (5.15 ± 0.05 U/(g·min·FW)) was higher than that of other groups (4.09 ± 0.07 to 4.48 ± 0.04 U/(g·min·FW)) (*p* < 0.05). After storage for eight days, the CAT activity of all groups decreased, but the CAT activity of the 5% O_2_/5% CO_2_/90% N_2_ group was higher than that of other groups (*p* < 0.05).

The above results show that 5% O_2_/5% CO_2_/90% N_2_ treatment is able to reduce the PPO activity, thus alleviating browning in fresh-cut *D. rubrovolvata* to the maximum extent in comparison with other treatments (Figure 4a). This result was consistent with the browning degree (Figure 3b). In our study, we found that the change in CAT activity was similar to that of the MDA content (Figure 2c); a 5% O_2_/5% CO_2_/90% N_2_ treatment could well delay the time of the CAT peak (Figure 4b).

#### 3.2.4. Changes in Nutrition

Total phenolic, polysaccharides, and free amino acids are important nutritional indicators in *D. rubrovolvata* [4,5,6]. The nutritional effect of different CO_2_ treatments (0%, 5%, 10%, 15%, or 20%) on fresh-cut *D. rubrovolvata* was investigated (Figure 5). The total phenolic content of all samples decreased during the shelf life of eight days (Figure 5a). In spite of this, it is worth noting that total phenolic levels in fruiting bodies treated with 5% O_2_/5% CO_2_/90% N_2_ were higher than in other gas environments (*p* < 0.05). On the eighth day of storage, the total phenolic content of the sample under the condition of 5% O_2_/5% CO_2_/90% N_2_ was 59.19 ± 0.53 mg/g, while that of other treatment groups was 48.34 ± 0.60 to 56.80 ± 0.50 mg/g. Lower total phenolic levels were found in other treatments (8.01 to 18.33%) compared to that of 5% O_2_/5% CO_2_/90% N_2_ samples on the eighth day (*p* < 0.05).

A similar trend was observed for polysaccharides content in the samples (Figure 5b). During the shelf life, the lowest deterioration in polysaccharides was observed for 5% O_2_/5% CO_2_/90% N_2_ in fresh-cut *D. rubrovolvata* compared with other groups. In the six days of storage, the polysaccharides content of the 5% O_2_/15% CO_2_/80% N_2_ treatment group decreased slowly, then began to decline sharply after 6 d. The polysaccharides content of 5% O_2_/5% CO_2_/90% N_2_ decreased slowly during the shelf life. On the eighth day of storage, the polysaccharides content of 5% O_2_/5% CO_2_/90% N_2_ reached 0.37 ± 0.01 g/(100 g). This was significantly higher than in other treatments (0.30 ± 0.00 to 0.34 ± 0.01 g/(100 g)) (*p* < 0.05).

Unlike total phenolic and polysaccharides, free amino acids in the samples of each group first increased, then decreased (Figure 5c). On the eighth day, compared with other groups, the total content of free amino acids under 5% O_2_/5% CO_2_/90% N_2_ was 4.88 ± 0.07 mg/g, 0.77, 0.9, 1.3, and 1.14 mg/g higher than in fresh-cut *D. rubrovolvata*, respectively (*p* < 0.05).

Li et al. [27]. found that the high respiratory rate indicated that the respiratory metabolism was vigorous, hence the nutrients stored in edible fungus would be rapidly consumed, leading to a rapid decline in storage quality. This result was coincident with our work; 5% O_2_/5% CO_2_/90% N_2_ could significantly inhibit the increase of the respiratory rate of fruiting bodies (Figure 2b) and reduce the consumption of nutrients in samples.

#### 3.2.5. Changes in Umami

5′-nucleotides, flavor amino acids (bitter, sweet, umami), and EUC can be used to assess the taste of edible fungus [29]. The effect of different CO_2_ treatments (0%, 5%, 10%, 15%, or 20%) on the umami taste of fresh-cut *D. rubrovolvata* was investigated (Figure 6). Nucleotides are crucial to the umami taste of edible fungi. As shown in Figure 6a, all fresh-cut fruiting bodies contained four 5′-nucleotides (5′-GMP, 5′-AMP, 5′-IMP and 5′-XMP) in the first four days, which impacted the umami taste of mushrooms. However, only three nucleotides were detected in the 0%, 5%, and 10% CO_2_ environments, and only two nucleotides were detected in the 15% and 20% CO_2_ environments on the eighth day of storage. Compared to other groups, the fruiting bodies from 5% O_2_/5% CO_2_/90% N_2_ environments had significantly higher contents of 5′-nucleotides (*p* < 0.05), with values of 0.58 ± 0.02 g/kg FW and 0.32 ± 0.01 g/kg FW after eight days as compared to 0.17 ± 0.00 to 0.42 ± 0.02 g/kg FW and 0.17 ± 0.00 to 0.29 ± 0.01 g/kg FW in other groups.

In addition, flavor amino acids played a crucial role in the fruiting body taste. Among these, only umami amino acids such as aspartic (Asp) and glutamic (Glu) acids contribute to the characteristic umami taste [38]. The Asp and Glu contents in the 5% O_2_/5% CO_2_/90% N_2_ treatment group were 0.10 ± 0.01 mg/g and 1.13 ± 0.03 mg/g, higher than in other gas environments by 111.11 to 142.86% and 100.89 to 129.89% on the eighth day, respectively (*p* < 0.05) (Figure 6b).

The combined action of 5′-nucleotides and umami amino acids (Asp and Glu) enhanced the umami flavor of the edible fungi [29]. In general, EUC represents their synergy. According to Figure 6c, the EUC values of a sample under a 5% O_2_/5% CO_2_/90% N_2_ gas environment (9.43 ± 0.33 g MSG/kg) at eight days are significantly higher than those observed in other storage environments (4.79 ± 0.17 to 7.96 ± 0.11 g MSG/kg), and approximately 1.18 to 1.97 times higher than those of the fruiting bodies (*p* < 0.05).

#### 3.2.6. E-Nose Analysis

The E-nose is sensitive to odor information obtained from the food samples, and slight changes in volatile compounds will cause different sensor responses. A radar chart was used for the statistical analysis to illustrate relationships and trends among variables. There was consistency in the signal values of the appearance change of 10 sensors during storage in these samples. However, significant differences were observed not only between the radar profiles of the samples of *D rubrovolvata* on days 0 and 8, but also between the edible fungi treated with different treatments, indicating that the volatile compounds in the samples changed significantly during the shelf life (*p* < 0.05). On the eighth day, the response value profile of the 5% O_2_/5% CO_2_/90% N_2_ treatment group was the most similar to 0 d. This indicated that the volatile components were the closest to the initial level (Figure 7a). With the increase in CO_2_ concentrations (10%, 15%, or 20%) in the storage environment, there was an obvious deviation between the response value profile of the samples and initial value. Figure 7b shows the results of the PCA analysis. On the eighth day, the distribution of *D. rubrovolvata* compared with the initial day could be observed by PCA under various conditions. The variance contribution rates of PC1 and PC2 were 68.6% and 18.9%, respectively. The first two PCs were 87.50% (more than 85%), which could determine most of the volatile substance information of the sample. At 8 d, the fruiting body under a 5% O_2_/5% CO_2_/90% N_2_ environment compared with other samples was closest to initial level, and with an increase in the CO_2_ concentration in the storage environment, the difference from initial level was increased. As shown in Figure 6c and Figure 7, compared with other treatments, 5% O_2_/5% CO_2_/90% N_2_ not only delayed the decline of sample EUC, but also preserved its odor.

#### 3.2.7. Changes in Total Colony Numbers

In general, CA with 2–5% O_2_ or CO_2_ was applied for fresh-cut products, which reduced the respiration rate and the growth rate of microorganisms [39]. Waghmare et al. [40] found that low O_2_ combined with a suitable storage environment for CO_2_ could effectively inhibit the growth of microorganisms in fresh-cut coriander. Consistent with the trend in the decay rate and respiration rate (Figure 2a,b), the total colony numbers of fresh-cut *D. rubrovolvata* gradually increased during the different gas environments for 8 d (Figure 8). Different treatments resulted in different rates of microbial growth. Interestingly, no microbial growth was observed in fresh-cut fruiting bodies after storage for 4 d at a CA of 5% O_2_/5% CO_2_/90% N_2_, whereas microbial growth was observed in the other treatment groups (*p* < 0.05). The total colony numbers were significantly lower (*p* < 0.05) in the fruiting body at 5% O_2_/5% CO_2_/90% N_2_ than the other groups, with values of 950 ± 141.42 CFU/g after eight days as compared to 2000 ± 141.42 to 5750 ± 212.13 CFU/g in other environments.

## 4. Conclusions

*D. rubrovolvata* is prone to problems such as the rapid softening and decay of fruiting bodies after harvest. According to the results of the experiment, we found that fresh-cut *D. rubrovolvata* was very sensitive to the concentration of O_2_ and CO_2_. Low O_2_ (5%) combined with an appropriate concentration of CO_2_ (5%) would be beneficial to the storage of fresh-cut *D. rubrovolvata*. Compared with other groups, 5% O_2_/5% CO_2_/90% N_2_ could maintain water in the cells of fresh-cut *D. rubrovolvata* and inhibit the activity of PPO and CAT. Thus, it could effectively inhibit a rise in the decay rate, respiratory rate, relative conductivity, MDA content, ethanol content, shear force, browning degree, the total number of colonies, and retard the decline of total phenolic, polysaccharides, free amino acids, and EUC. The flavor of the sample could also be well maintained. In conclusion, the storage environment of 5% O_2_/5% CO_2_/90% N_2_ could significantly delay the deterioration in the quality of *D. rubrovolvata* during the shelf life and have high food safety. Therefore, a CA with optimal O_2_/CO_2_/N_2_ concentration could be a recommended technical method in other fresh-cut edible fungi.

## Figures and Tables

**Figure 1 foods-12-01665-f001:**
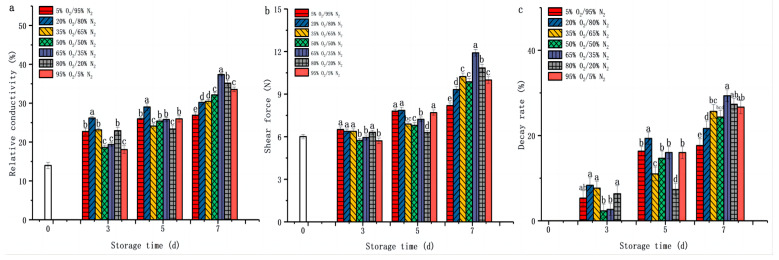
Effect of O_2_ concentration (%) on relative conductivity (**a**), shear force (**b**), and decay rate (**c**) of fresh-cut *Dictyophora rubrovolvata*. Differences at *p* < 0.05 are indicated by different letters and data are expressed as the mean ± SD.

**Figure 2 foods-12-01665-f002:**
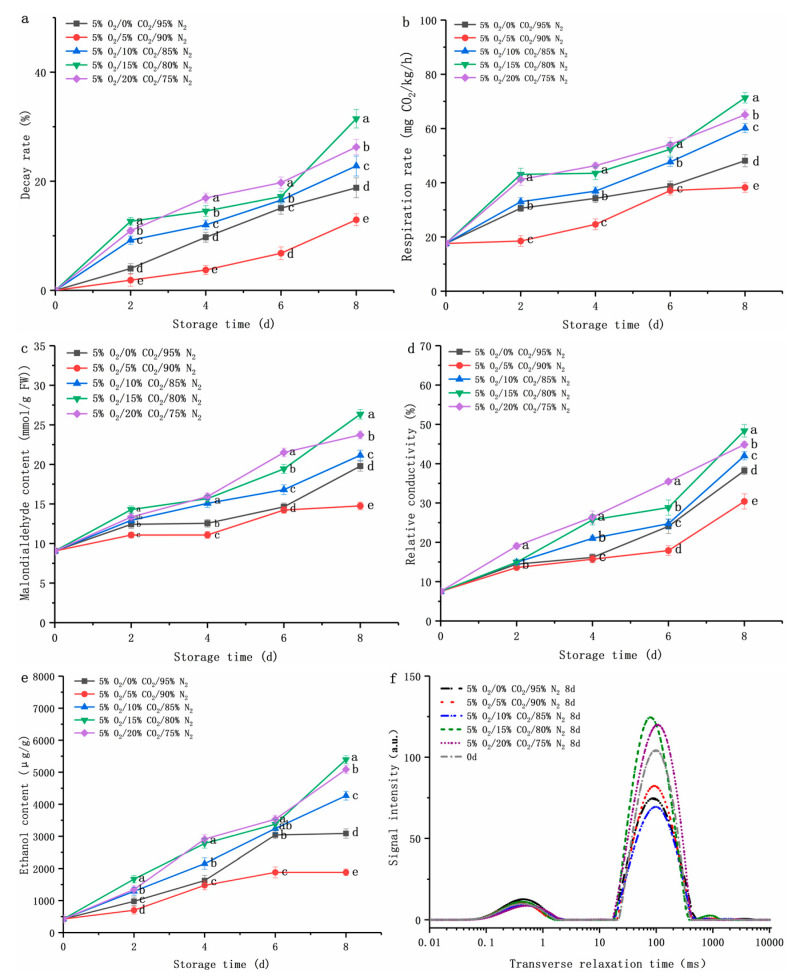
Effect of CO_2_ concentration (%) with 5% O_2_ on decay rate (**a**), respiration rate (**b**), malondialdehyde content (**c**), relative conductivity (**d**), ethanol content (**e**), and transverse relaxation time (**f**) of fresh-cut *Dictyophora rubrovolvata*. Differences at *p* < 0.05 are indicated by different letters and data are expressed as the mean ± SD.

**Figure 3 foods-12-01665-f003:**
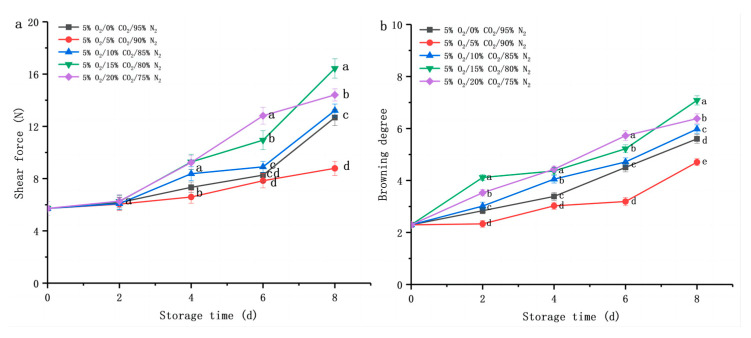
Effect of CO_2_ concentration (%) with 5% O_2_ on shear force (**a**), and browning degree (**b**) of fresh-cut *Dictyophora rubrovolvata*. Differences at *p* < 0.05 are indicated by different letters and data are expressed as the mean ± SD.

**Figure 4 foods-12-01665-f004:**
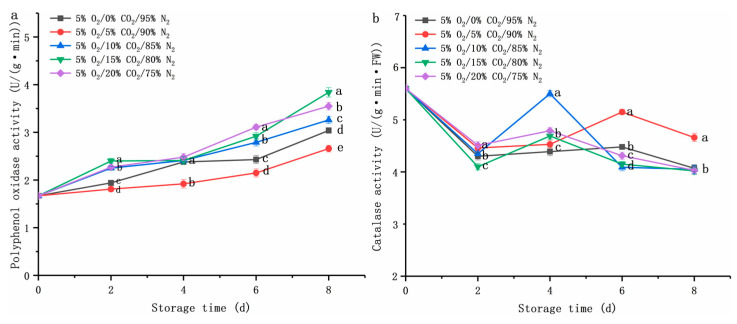
Effect of CO_2_ concentration (%) with 5% O_2_ on polyphenol oxidase activity (**a**), and catalase activity (**b**) of fresh-cut *Dictyophora rubrovolvata*. Differences at *p* < 0.05 are indicated by different letters and data are expressed as the mean ± SD.

**Figure 5 foods-12-01665-f005:**
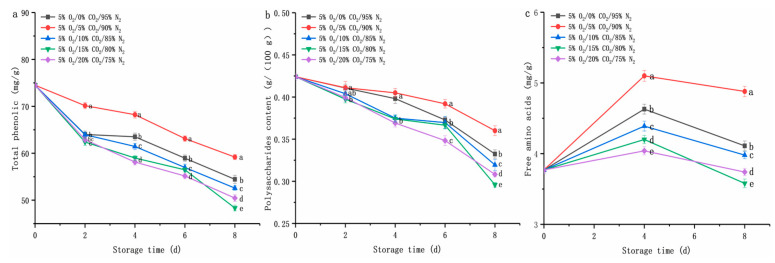
Effect of CO_2_ concentration (%) with 5% O_2_ on total phenolic (**a**), polysaccharides content (**b**), and free amino acids (**c**) of fresh-cut *Dictyophora rubrovolvata*. Differences at *p* < 0.05 are indicated by different letters and data are expressed as the mean ± SD.

**Figure 6 foods-12-01665-f006:**
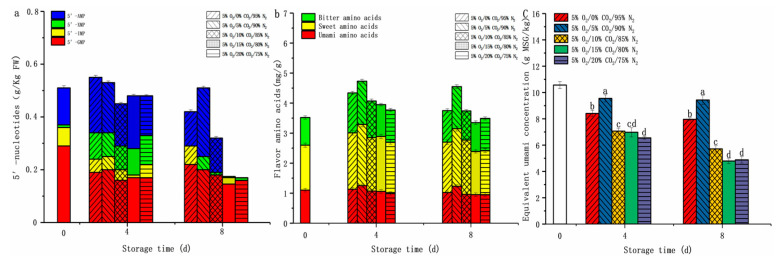
Effect of CO_2_ concentration (%) with 5% O_2_ on 5′-nucleotides (**a**), flavor amino acids (**b**), and equivalent umami concentration (**c**) of fresh-cut *Dictyophora rubrovolvata*. Differences at *p* < 0.05 are indicated by different letters and data are expressed as the mean ± SD.

**Figure 7 foods-12-01665-f007:**
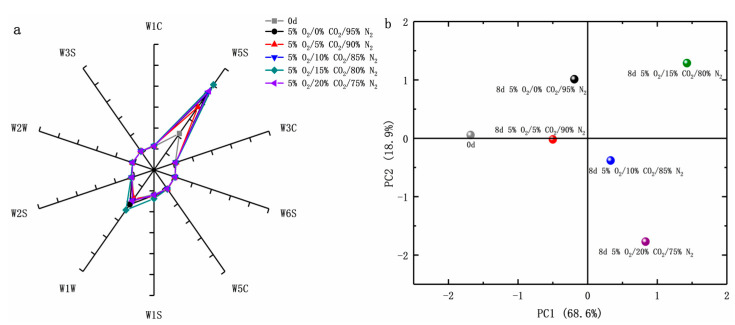
Effect of CO_2_ concentration (%) with 5% O_2_ on radar fingerprint charts (**a**), and principal component analysis (**b**) of fresh-cut *Dictyophora rubrovolvata*. Differences at *p* < 0.05 are indicated by different letters and data are expressed as the mean ± SD.

**Figure 8 foods-12-01665-f008:**
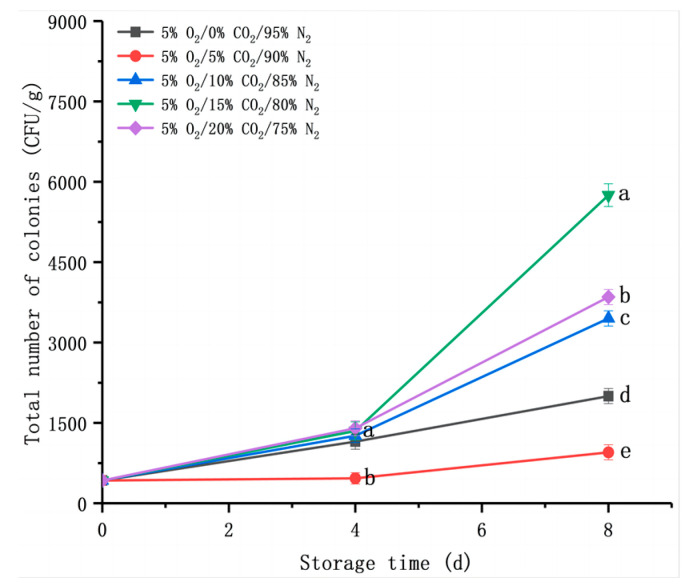
Effect of CO_2_ concentration (%) with 5% O_2_ on total colony numbers of fresh-cut *Dictyophora rubrovolvata*. Differences at *p* < 0.05 are indicated by different letters and data are expressed as the mean ± SD.

## Data Availability

Not applicable.

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
