# Peer review of "Effect of Controlled Atmosphere Packaging on the Physiology and Quality of Fresh-Cut Dictyophora rubrovolvata"

_foods, 2023, doi:10.3390/foods12081665_

Round 1
Reviewer 1 Report
The manuscript entitled: "Study and Application of Controlled Atmosphere Packaging on Fresh-cut Dictyophora rubrovolvat" is about the application of CAP for a fresh-cut edible fungus from China. In general, the manuscript is interesting and has enough novelty. The methodology is covering the objectives of the research. The objectives of the research well fit the journal's aims and scopes. There are some comments that need to address carefully by the authors before the final decision by the editor:
1- Title: The title is OK, and you can improve it; for example instead of "Study and Application" better to use study or application. or you can find another informative title to catch the eyes of the readers.
2- Abstract: Almost OK, support the results with some quantitative data.
3- Keywords: Choose keywords other than the main words in the title. It will improve the visibility of the article.
4- Introduction: It is well developed and needs to improve the literature background by bringing other methods used for fresh fruits and vegetables. For example, this is a good example of a reference that may improve the background: Esmaeili, Yasaman, et al. "The synergistic effects of aloe vera gel and modified atmosphere packaging on the quality of strawberry fruit." Journal of Food Processing and Preservation 45.12 (2021): e16003.
5- Novelty statement: Improve this part at the end of the introduction before the objectives. Also, try to use passive tense instead of active [Line 98]. Do not write such sentences in the introduction that shows you are reporting results [Lines 98-99].
6- Materials: Separate the materials including chemicals in a new subheading.
7- Methods: All methods must have a proper reference(s) either a published article or standard method.
8- Results and discussions: This part is ok and well-developed.
9- Conclusion: This part is too long, make it short. Focus on hypothesis justification and research recommendation.
Author Response
Sincerely thank you for taking time out of your busy schedule to review my manuscript and provide valuable suggestions. We have revised it carefully, please see the attachment.

Reviewer 2 Report
Dear Editor, in the manuscript Foods-2324574 authors evaluated effects of packaging in atmospheres with different O2 and CO2 concentration on quality traits of fresh cut Dictyophora rubrovolvata. The manuscript shows new and interesting information because the literature about storage technologies for this fungus is scarce. The following comments should be considered before acceptance:
- Usually, the storage of fruit and vegetables in plastic packages with a particular O2 and CO2 composition is named as active packaging instead of controlled atmosphere. This issue should be considered in the tittle and in the whole manuscript.
- Line 23: Delete brackets.
- Lines 133: How many bags were taken for each treatment and sampling date to perform all the analytical determinations? This is an important issue that should be clarified in the revised manuscript.
- Line 158: Add full name for LF-NMR.
- Lines 167, 174, etc: Use g instead of rpm for centrifugation speed.
- Line 304: In this sentence, reference number 37 is cited. However, reference number 36 has not been cited before. Check the manuscript and correct the reference number properly.
- 3.2.1 section: Discussion for these physicochemical parameters should be improved because this is limited to say that previous papers showed “the same results”
- Line 228: “Not the same” because the best composition in the present results was 5% O2+ 5% CO2.
- Line 347: Degradation of cell wall components and structure usually leads to firmness losses. However, in the present experiments shear force increased showing a hardening increase of fungi tissues. This issue should be explained and discussed.
- Line 379: However, the present results show higher PPO as CO2 concentration increased. An explanation to these contradictory results should be addressed.
- Lines 464-474: Figure 7a should be cited in this paragraph.
- 3.2.7 section: Discussion for this section should be provided.
- Conclusion is too long and should be shortened by focusing just in the most relevant results and conclusions.
- Results of the first experiment with 5% O2 treatment should be compared with those of the second experiment with 5% O2+5% CO2 treatment in order to point out if the combination of 5% O2 plus 5% CO2 lead to better results in terms of quality maintenance than just 5% O2 treatment.
- The meaning of letters should be added to all the figure legends.
Author Response

(The authors gave the same response as above.)

Reviewer 3 Report
Foods-2324574 Study and Application of Controlled Atmosphere Packaging on Fresh-cut Dictyophora rubrovolvata
The manuscript presents the complex issue of how to keep the edible mushroom Dictyphora rubrovolvata as long as possible. The authors try to find the optimal composition of the protective atmosphere in order to achieve the highest quality product. Quality is measured by a number of parameters: texture, degree of browning, nutritional composition, content of components that make up the so-called umami, volatile components and the total number of microorganisms. Furthermore, the contents of PPO enzymes and CAT profile are determined as part of the quality.
I have some comments given in the following list.
- General comment: there are frequently applied number of reference very far from the first author name, see e.g. lines 81-83, 89-91, 153-154, 187-188, 228-229, 231-232, 340-338, 378-379, 425-427.
- - General comment: The legend in all the figures is difficult to read when the manuscript is printed on paper.
- - Line 117 “stalks were cut into three sections”. There is no detail how stalks were cut. Perpendicular to the body axis?
- - Lines 118-124 this text is part of Material and methods chapter but relates on results apparent from Figure 1.
-- Lines 153-154 “describe” how relative conductivity was measured. Reader needs to see reference to be able to recognize that conductivity is electrical, not thermal. I recommend using here the word “electrical” and briefly describing method used: voltage, direct or alternating current, frequency.
-- Line 184 “Polysaccharides were not was”.
- - Lines 234-236 there is missing number of replications of all measurements.
- - Line 240 input “electrical” between relative and conductivity.
- - Line 344 unit of texture should be using SI standard. Instead gram should be used Newton. This is valid also for Figure 3 a).
Author Response

(The authors gave the same response as above.)
